# Analysis of the Ecological Carrying Capacity of Fish Resources in Shengjin Lake, Anhui Province, China

**DOI:** 10.3390/ijerph19138177

**Published:** 2022-07-04

**Authors:** Guiyou Zhang, Yan Lu, Zijun Fang, Hong Yang, Zhong Wei

**Affiliations:** 1College of Economics & Management, Anhui Agricultural University, Hefei 230036, China; luyan@ahau.edu.cn (Y.L.); 21721370@stu.ahau.edu.cn (Z.F.); yanghong666@stu.ahau.edu.cn (H.Y.); 2College of Animal Science and Technology, Anhui Agricultural University, Hefei 230036, China; wz6565@ahau.edu.cn

**Keywords:** fish resources, carrying capacity, Shengjin Lake

## Abstract

The carrying capacity of fish is related to the sustainability of fisheries’ activities in water bodies. The fish carrying capacity of a water body is the maximum fish yield that can be carried by the natural bait organisms in the water body under the ideal natural conditions without feeding and fertilization. The evaluation of fish productivity is an important basis for rational stocking, rational fishing, and the scientific utilization of natural bait resources in a water area. This paper adopts the background data of the Shengjin Lake fishery ecosystem and uses the bait-based estimation method. The results show that (1) the annual yield of silver carp fed on phytoplankton is 1.5 million kg; (2) the annual yield of bighead carp fed on zooplankton is 1.295 million kg; (3) the annual yield of benthos is 310,000 kg; (4) the annual yield of organic detritus is 280,000 kg; and (5) as the coverage of water grass in Shengjin Lake is less than 10%, it should be protected and restored rather than used by fish. In general, the annual maximum carrying capacity of fish in Shengjin Lake is 3.385 million kg, except for water and grass.

## 1. Introduction

Large water surfaces, such as lakes and reservoirs, are major parts of China’s inland fishery areas. The large water surface fishery industry is a main part of China’s freshwater fisheries, playing an important role in aspects of building ecological culture in water areas, ensuring the supplementation of high-quality aquatic products, promoting industry integration, and motivating the increases in the income of fisherfolk. In recent years, along with the vigorous expansion of fisheries, people have acquired more fishery knowledge and gradually realized that fish resources are renewable but limited; therefore, it is necessary to properly exploit, protect, and manage fish resources [1]. Carrying capacity is a concept closely related to resource endowment, technical means, social choices and values, limit connotations, and ethical characteristics, in addition to being uncertain and nonlinear [2,3]. The carrying capacity is the upper limit within a certain ecological fitness. It refers to the maximum abundance of a species that can be sustained within a given area of a habitat. In terms of application, it can bee classified into biology and ecology, applied ecology, population ecology, and other perspectives [4]; it can be based on the research methods studied according to special empirical methods, general empirical methods, and theoretical methods; and, practically, it mainly covers the population or economy carrying capacity of natural resources or the environment, the carrying capacity of a specific biological population in a habitat, and the tourist carrying capacity of entertainment and tourism systems [5,6,7,8,9]. Therefore, the concept of the carrying capacity of resources and the environment has gradually become more comprehensive and systematic.

The fish carrying capacity of a water body determines the sustainability of fishery activities. The United States, the United Kingdom, Australia, and Canada were the first countries and regions to implement the sustainable utilization of fish resources. In 1954, Gordon proposed the concept of the maximum economic yield (MEY) [10]. Later, many scholars put forward theories and research methods of fish resource measurement from the perspectives of economy and society based on the biological characteristics of fish resources, which provide policy and theoretical foundations for the sustainability of fish resources [11,12,13,14,15,16,17,18,19]. Since 1998, the UN Food and Agriculture Organization has conducted extensive research on the index system and measurement of the sustainable utilization of fish resources, such as the FAO Indicators for *Sustainable Development of Fisheries* [20], the *FAO Indicators for Sustainable Development of Marine Capture Fisheries* [21], and the *Fisheries Sustainability Indicators: The OECD Experience by the Organization for Economic Co-operation and Development* [22].

However, compared with developed countries, China still lags behind in the research on fish resource utilization, management, and evaluation. In 1995, Professor Zhang Xiangguo translated the *Bioeconomic Analysis of Fisheries*, a monograph by the Norwegian scholar Hannesson [23]. In addition, some scholars have evaluated the significant traditional commercial fish resources in offshore areas of China. For example, Zhan Bingyi evaluated the allowable catch and maximum economic yield of northern filefish as well as its development based on a bioeconomic decision-making model [24]. Since 2000, Chen Xinjun and other scholars have studied the systematic evaluation of the sustainable utilization of fish resources, the BP-model-based comprehensive and dynamic evaluation of the sustainable utilization of fish resources, the grey relative correlation-based evaluation of the sustainable utilization of fish resources, and other evaluation and measurement methods for the sustainable utilization of fish resources [25,26,27]. The Gu Xiaohong and Mao Zhigang research group of the State Key Laboratory on Lakes and Environment of Nanjing Geography and the Lakes Research Institute of CAS conducted extensive and in-depth research on the fish community structures of large water surfaces and the change in fish resources. They successively investigated the fish community structure and fish resources of Taihu Lake, Hulun Lake, and Hongze Lake and analyzed the fish community structures and resource change trends in the above water bodies based on historical statistical data [28,29,30,31]. The above research provides important references for this paper.

As one of the lakes flowing into the middle and lower reaches of the Yangtze River, Shengjin Lake is a major overwintering place and ceasing place for East Asian and Australian migratory water birds. It has a significant ecological status and international influence and was approved as a National Natural Reserve in 1997. The water level of Shengjin Lake decreases in autumn and winter and increases in spring and summer with the change in seasonal precipitation, and it has a higher fish production capacity [32,33]. However, with the increasing environmental constraints in recent years, the fishery development space of Shengjin Lake has greatly shrunk, such that the fish production management method requires an urgent transformation as well as upgrading. In the face of the new situation, and according to the demands of a healthy ecologic system of large water surface and fishery development, how to conduct regulatory activities of fish production in Shengjin Lake and how to promote the harmonious development of ecology, production, and life of the water area have become real problems that urgently need to be solved. This paper follows the principle of sustainable utilization of fish resources and uses the background data of the Shengjin Lake fishery ecosystem to evaluate the fish production potential of Shengjin Lake. The research objective is to scientifically set sustainable fishing quotas to optimize fishing.

## 2. Profile of the Research Area

Shengjin Lake is located at the riverside of the junction between Dongzhi County and Guichi District, Chizhou City, Anhui Province. The area of the lake is 13,280 hm^2^. There is a provincial hydrometric station named the Huangpenhe Hydrometric Station in the Shengjin Lake National Natural Reserve (116°55′–117°15′ E, 30°15′–30°30′ N), from which it is possible to realize the real-time monitoring of the water regimen data of Shengjin Lake.

### 2.1. Fish Resources in Shengjin Lake

In order to research an innovative model of the sustainable development of lakes in the middle and lower reaches of the Yangtze River, in July and October 2012, as well as January and April 2013, the National Freshwater Fishery Engineering Technology Research Center (Wuhan) of the Institute of Hydrobiology (IHB), Chinese Academy of Sciences (CAS), in cooperation with the School of Fisheries, Huazhong Agricultural University, conducted an investigation on the biological resources in Shengjin Lake. The contents of the investigation contained phytoplankton, zooplankton, benthic organisms, aquatic vascular plants, and fish; in which, for investigation of the fish, aside from the four seasonal investigations, specific investigations were conducted in the fishing period of spring, autumn, and winter. The data on the biological resources of Shengjin Lake in this paper are based on the historical data collected through a literature review. Meanwhile, from July 2016 to June 2017, the research group filed an investigation of the Shengjin Lake National Natural Reserve every month, and they visited the local fishery and aquatic departments to review the fishery history and aquatic records. In order to investigate the status of fish diversity, they collected and investigated the quantities and varieties of fish every two months in a fixed market for fish; made evaluations of the status of the habitat for key species and took corresponding protections; discovered the spawning periods and spawning grounds of fish by collecting samples and visiting fisherfolk; analyzed the features of spawning grounds and their distributions; and made evaluations of the diversity of fish to analyze the number of species, the constitutions of economic fish, fish age structures, and the areas of spawning grounds (see Figure 1).

There are 66 fish species of 19 families in Shengjin Lake, where: ① Cyprinidae fish, up to 40 species, account for more than 60% of the total number of species; ② small fish are numerous, both in the number of species and the population size; ③ most of the fish belong to settled ecological groups, adaptive to the lentic environment of the lake; and ④ due to the control gates, fish resources, such as black carp, grass carp, silver carp, and bighead carp, which were originally migratory fishes in the rivers and lakes, cannot be naturally supplied, but are maintained through breeding by human beings. There are many fish species with fishery significance but a low yield in Shengjin Lake, mainly including silver carp, bighead carp, crucian carp, grass carp, *Megalobrama skolkovii*, *Pelteobagrus fulvidraco*, mandarin fish, snakehead, and Culter.

### 2.2. Fishery Habitat of Shengjin Lake

Most of the fishes in Shengjin Lake are oviposition-producing fishes, and the aquatic vegetation provides an ideal natural spawning ground for them, including two types: one is clusters of submerged plant communities, such as those near Yangetou, at the junction of the Zhangxi River mouth; the other is marsh plant communities at the lakeside or river mouth, distributed around Babaizhang. Through interviews with fishermen and data enquiries, the research team learned that there are two large spawning grounds in the lake (see Table 1).

Shengjin Lake is an ephemeral lake. During the winter, as a result of the opening of Huangpen Gate, the water level and water area of Shengjin Lake are greatly reduced, especially in the upper lake. The wintering areas in the upper lake are mainly in the vicinity of the Jiahe River. There is no fixed wintering area in the middle and lower lakes, and the deep-water area can provide a wintering habitat for fish.

There is no large-scale fish feeding ground in Shengjin Lake, but all areas with rich fish feed in the lake can be feeding habitats for fish, and most spawning and wintering areas are feeding areas as well. The feeding areas for carp, crucian carp, gobio, rhodeinae, snakehead, Xenocypris, catfish, etc., are mainly in areas with a slow water flow.

## 3. Data and Research Methods

The fish carrying capacity of a water body, also known as the potential for fish production, refers to the maximum fish production that natural food organisms in the water body may carry under ideal natural conditions, without artificial feeding or fertilization. As a consequence, fish productivity is a kind of potential and is also the capacity to convert varieties of organisms, including nonorganic and organic nutrients, in water into fish products under specific circumstances. The actual fish productivity refers to the maximum fish yield under current conditions in a water body, and the evaluation of fish productivity is considered an important basis for rational breeding and fishing, as well as for the scientific utilization of natural food resources in waters.

Here, a few points need to be explained. First, in the process of this study, the research group strictly abides by the implementation rules of the Fisheries Law of the People’s Republic of China, the Yangtze River fisheries resources management regulations, and the wetland protection regulations of Anhui Province. Secondly, the research only collected relevant data and did not use fish or any other animals for experiments, so there were no cases of fish or other animals being handled. Thirdly, the researchers told all of the interviewees the purpose of collecting the survey data; that is to say, the interviewees knew the purpose of the interview.

### 3.1. Data Collection on Aquatic Plants, Phytoplankton, Zooplankton, and Benthic Animals

#### 3.1.1. Aquatic Plants

The natural distribution of aquatic plants is closely related to the depth, transparency, and substrate of water. Generally, there are rich species of aquatic plant communities in highly transparent shallow water, at the bottom of which is more humic sludge, while aquatic plants are sparsely distributed in deep waters or waters with a sandy bed. In larger deep ponds or lakes, there is a regular ring-shaped distribution; that is, there is an emergent aquatic plant belt, floating-leaved aquatic plant belt, and submerged aquatic plant from the shallow water area along the shore to the deep-water area in the center. Aquatic plants, especially planktonic algae, as good feeds for fish, also play an important role in purifying the ecological environment of the water. Aquatic vegetation can be divided into emergent, floating-leaved, and submerged types in terms of their ecological characteristics. However, in this article, hygrophytes on tidal flats are also included in the scope of aquatic vegetation. Therefore, there are, in total, four types of aquatic vegetation in the Shengjin Lake reserve.

Statistics on the biomass of aquatic plants can be counted in case conditions, and aquatic plants have a high density. Large aquatic plants, which are unevenly distributed, may be sampled based on points in dense areas, general areas, and sparse areas of each zone. The sampling point may be more or less, which can be determined according to the needs and possibilities. In addition, coverage, as an aspect of biomass, can be used as reference data for biomass.

The research group recorded the biomass of typical hygrophytes on tidal flats, emergent plants, floating-leaved plants, and submerged plants based on the wet weight of plants by the research group. Specifically, a sample size of 1 × 1 m was generated nine times by using simple quantitative sampling in a 1 m^2^ sampling frame, and then the wet weight was measured on-site. Then, the collected data were converted into the biomass per unit area (g/m^2^) and the total biomass. The main species surveyed were wild carrot and Shechuangzi of the umbellifer family; water chestnut of trapaceae and *Artemisia selegensis* of the composite family; curled pondweed of the Potamogetonaceae family and whorled water-milfoil of the haloragaceae family; hydrilla of the hydrocharitaceae family; duckweed of the lemnaceae family; hornwort of the ceratophyllaceae family; manchurian wild rice and stinkgrass of the poaceae family; and *Carex thunbergii* of the cyperaceae family. The results are shown in Table 2.

#### 3.1.2. Phytoplankton

Qualitative samples were collected by plankton net No. 25 in the pattern of “∞” at 30 cm below the water surface for 10 min and then fixed with 1.5% Lugol’s solution immediately for the qualitative analysis of phytoplankton. Quantitative samples were collected by drawing 10 L of water separately from the surface layer (10–20 cm) and the bottom layer (30–50 cm above the substrate sludge). After mixing, take 1 L of water and pour 15 mL of Lugol’s solution into the 1 L mixture for further use; the fixative solution was poured into a 1000 mL precipitator and allowed to stand for more than 24 h, after which the supernatant was slowly aspirated by the siphon method, and the remaining 30–50 mL precipitation was poured into a quantitative bottle and fixed to a quantity of 10 mL after adding 1–2 mL of a formaldehyde fixative for the quantitative analysis of phytoplankton.

In November 2016, a total of 55 species, 36 genera, and 6 phyla of phytoplankton were surveyed, among which species of *Bacillariophyta*, *Cyanophyta*, and *Chlorophyta* were the most common, while those of *Chrysophyta*, *Cryptophyta*, and *Euglenophyta* were less common. A total of 10 genera of *Bacillariophyta*, 13 genera of *Chlorophyta*, 7 genera of *Cyanophyta*, 2 genera of *Euglenophyta*, 2 genera of *Cryptophyta*, and 2 genera of *Chrysophyta* were detected. In general, under quantitative detection, the number of species of *Chlorophyta* was the largest, accounting for 41.82%, and they were the dominant population, followed by species of *Bacillariophyta*, accounting for 23.64%, and the number of the species of *Chrysophyta* were the least common, only accounting for 3.64%. Based on the principle that species with a Y-value of over 0.02 are dominant species, it was calculated that the main dominant species were *Melosira granulata* var. angustissima, *Synedra acusvar* and *Synedra ulna*, of *Bacillariophyta*, *Rhadbogloea smithii* and *Merismopedia marssonii* of *Cyanophyta*, and *Kirchneriella obesa*, *Schroederia spiralis*, and *Selenastrum gracile* of *Chlorophyta*. According to the results, the biomass, abundance, and diversity indexes of the middle lake were relatively higher than those of other lake areas, mainly because the middle lake is an important breeding base for the fishery and the feeds released in the middle lake made its water more nutritious.

#### 3.1.3. Zooplankter

Samples were collected by plankton net No. 13 in the pattern of “∞” at 30 cm below the water surface for 10 min and then fixed with a proper amount of formaldehyde solution immediately for the qualitative analysis of zooplankton. Qualitative and quantitative samples of rotifers were shared with that of phytoplankton. Qualitative samples of large macrozooplankton were obtained by plankton net No. 13, while quantitative samples were collected by drawing 5 L of water separately from the surface layer (30 cm) and bottom layer (30–50 cm above the substrate sludge), filtered, concentrated to 100 mL by net No. 25, fixed by adding 5% formaldehyde fixative for further use, and kept through precise precipitation and concentration for quantitative analysis of zooplankton.

Through the investigation in November, a total of 10 families of zooplankton, 13 genus and 10 species were detected. Among them, the species of *Brachionus* were the most common, totaling 5 species, followed by 4 species of *Trichocerca*, 2 species of *Keratella*, and 1 species of *Schizocera*, *Polyarthra*, *Asplanchna*, *Filinia*, *Rotaria*, *Diaphanosoma*, *Bosmina*, *Microcyclops*, *Thermocyclops*, and *Sinocalanus*, respectively. A total of nine dominant species were identified, including five species of *Rotifera*, two species of *Cladocera*, and two species of *Copepods*. Specific species and the dominance of the dominant species are shown in Table 3. The nauplii and copepodites were found throughout the lake.

In the survey, the density and biomass of zooplankton varied each month. In the first survey in August 2016, the average density of zooplankton in the whole lake was 246.57 ind./L, and the average biomass was 0.84 mg/L.

In the second survey in November, the average density of zooplankton in the whole lake was 468.91 ind./L, and the average biomass was 1.26 mg/L.

In the survey in August 2016, the average Shannon–Wiener diversity index of zooplankton in Shengjin Lake was 2.77, the Margalef species richness index was 1.81, and the Pielou evenness index was 0.71; in the survey in November, the average Shannon–Wiener diversity index for zooplankton in Shengjin Lake was 2.82, the Margalef richness index was 1.48, and the Pielou evenness index was 0.76.

#### 3.1.4. Benthic Animals

The collection process of a quantitative sample of benthic animals is carried out as follows: collect two effective samples per point by using a mud collector of 1/16 m^2^ and pour the mud sample collected into plastic basins, strain off the sludge with a 40 mesh sampling metal sieve on-site, collect the remainder in preservative plastic bags or sample bottles depending on the number of sample bottles, and then bring them back to the laboratory for sorting, counting, weighing, and species identification. In terms of the sample treatment method, species of Tubificidae were fixed and kept in sample bottles with a 5% formaldehyde solution, and other benthic animals were fixed and stored with 75% alcohol. The samples collected were converted into population density and biomass per unit area according to the methods of the Specifications on Inland Water Fish Resources Survey; the qualitative samples were based on the types of quantitative samples and were collected from time to time in any part of the lake, and, after collection, the detected samples were numbered, fixed, properly kept, and brought back to the laboratory for identification in a timely fashion.

Through anniversary qualitative and quantitative surveys of benthic animals in Shengjin Lake, a total of 25 taxa were identified, under 3 phyla, 4 classes, 6 orders, and 9 families. Among them, there were oligochaetas of two orders, two families, and five species, accounting for 20.0% of the total species; mollusks of three orders, three families, and five species accounting for 48%; and aquatic insects of one order, one family, and eight species accounting for 32.0%. The average density and average biomass of benthic animals in the whole of Shengjin Lake were 48.42 ind./m^2^ and 2.44 g/m^2^, respectively.

The Margalef and Shannon diversity indices of the whole of Shengjin Lake showed the same trend, being the highest in August and the lowest in November. The Margalef diversity index of the whole lake in August and November was 3.67 and 2.47, respectively, and the Shannon diversity index was 2.30 and 1.93, respectively. The Pielou evenness index and the diversity index of the whole of Shengjin Lake showed opposite trends: the evenness index was the highest in November and the lowest in August, 0.76 and 0.58, respectively.

### 3.2. Research Method

This paper used the bait-based estimation method to estimate the fish yield. The results of the bait-based estimation method could not only explain whether the ecosystem was overloaded but also make a more accurate judgment on the cause of overloading, which was conducive to taking intervention measures and countermeasures. 

#### 3.2.1. Estimate the Fish Yield According to Wafting Energy and Bait Base

The fish yield of herbivority, plankton feeding habits, and benthonic organisms are shown as Formula (1):F = B × *P/B* × a × V × S × 100/k(1)
where F is the fish yield provided by plankton, higher aquatic plants, and benthos; B is the annual mean biomass of plankton, higher aquatic plants, and benthos; *P/B* is the ratio of the annual yield of the bait to the average annual biomass; V is the reservoir capacity within 10 m of the surface layer of the water body; S is the fishing area; and k is the bait ratio of fish to the bait.

#### 3.2.2. Primary Productivity Estimation Method of Phytoplankton

Liang Yanling et al. (1988), on the basis of a large number of investigations on primary productivity and catch yield, explored the relationship between the two through mathematical statistics and obtained the regression equation. Based on the model to predict the fish yield in the condition that the primary productivity of phytoplankton in a certain water body was known [34]. The method focused on the primary productivity of phytoplankton, which was much more simple and clear, as shown in Formulas (2) and (3).
Y_G_ = 370X − 852(2)
Y_N_ = 364X − 970(3)
where Y is the fish yield (kg/ha), Y_G_ is the gross yield, Y_N_ is the net yield, and X is the DO detected by the light and dark bottle technique (mg/L).

#### 3.2.3. The Fish Yield Provided by Organic Detritus

The formula of the fish yield provided by organic detritus is as follows (4):F = CsV(19.6%A + 22.6%B) × 3.9 × 105/(3560A + 3350B)(4)
where F is the fish yield provided by organic detritus, Cs is the content of organic carbon in the organic detritus, A is the ratio of silver carp to silver carp and bighead carp in the water body, and B is the ratio of bighead carp to silver carp and bighead carp. Since only the maximum carrying capacity of a water body is considered in this paper, with the premise of no fertilization and bait, the bait-based estimation method is obviously the most ideal method.

## 4. Results and Analysis

### 4.1. Plankton-Based Fish Productivity

The fish productivity based on plankton is estimated according to the following Formula (5):Fy = Bp × *P/B* × Ur ÷ Fr × Fa × Cd(5)
where Fy is the fish yield, Bp is the biomass of plankton, Ur is the utilization rate, Fr is the feed ratio, Fa is the feeding area, and Cd is the compensation depth. The values of the *P/B* ratio, utilization, and feed ratio of the plankton were determined by referring to those of the East Lake in Wuchang City in the mid-China area, while the values of the feed ratio and other ratios were determined according to the specifications of the Evaluation Criteria of Fish Yield in Reservoir. For large water bodies, hypophthalmichthys molitrix (silver carp), a kind of filter-feeding fish, was the main fish fed on phytoplankton; therefore, the fish yield is based on the phytoplankton here was the carrying capacity of fish in Shengjin Lake. The area of the water environment suitable for fish activities is calculated as 100 square kilometers, and the compensation depth is 2 m. Based on the foregoing formula, the productivity of silver carp feeding on phytoplankton is 1.5 million kg/year. In large water bodies, as the fish feeding on zooplankton is mainly the filter-feeding fish silver carp (known as the bighead), the zooplankton-based fish productivity is the silver carp carrying capacity of Shengjin Lake. Based on the foregoing formula, the productivity of silver carp feeding on zooplankton is 12,950,000 kg/year. Figure 2 provides the trends in the zooplankton density and biomass in each lake area of Shengjin Lake. In general, the change trends in the zooplankton density and biomass are the same: the quantity is the most in November, followed by the quantity in August, and the quantity is the least in January.

### 4.2. Benthic-Animal-Based Fish Productivity

The fish productivity of benthic animals was calculated based on their fish capacity as fish food, which was determined through the energy estimation method. The calculation formulae for aquatic insects, aquatic oligochaetes, and mollusks are slightly different due to their biological characteristics (such as the *P/B*, the dry–wet weight ratio, energy values, etc.). Estimations were carried out according to the formula of Liang Yanling et al. (1998), as (6)–(8):

Estimation of the fish productivity of aquatic insects:FI = 0.183 BI(6)

Estimation of the fish productivity of aquatic oligochaetes:FO = 0.235 BO(7)

Estimation of the fish productivity of mollusks:FM = 0.032 BM(8)

The fish productivity of benthic animals was 310,000 kg. There are many fishes that feed on benthic animals, mainly demersal fish in shallow waters, such as black carp, carp, crucian carp, *Pelteobagrus fulvidraco*, and bitterlings. These fishes, though growing slower than silver carp and bighead carp and being smaller, are high-quality fish, favorable in price, and mostly settled fishes showing ecological diversity. Figure 3 provides the trends in the density and biomass of benthos in each lake area of Shengjin Lake. With the change in seasons, the density and biomass of the three lake areas of Shengjin Lake changed greatly. The density of Shengjin Lake and the upper lake is the highest in August, followed by March, and is the lowest in November. The density variation law of the middle lake and the lower lake is the highest in August, followed by November, and is the lowest in March. The biomass and density of Shengjin Lake showed similar seasonal changes.

### 4.3. Fish Productivity Supported by Organic Debris in the Water Body

Due to the failure to obtain accurate information on the organic debris of Shengjin Lake, the productivity of Xenocyprininae fish (mainly including *Xenocypris davidi* Bleeker and *Xenocypris microlepis*) can be estimated by empirical formulae. Generally, in large water body fisheries, the productivity of demersal fish feeding on organic debris generally accounts for about 10–15% of that of filter-feeding fish. Considering the great change in the water level of Shengjin Lake, it is based on 10% herein. Therefore, the productivity of Xenocyprininae fish feeding on organic debris is 280,000 kg/year.

### 4.4. Fish Productivity Based on Higher Aquatic Plants

The emergent plants, most of the floating-leaved plants, and the submerged plants in higher aquatic plants are not good food for fish, so only a small part of the submerged plants (such as *Spirodela polyrhiza* and hornwort) that fish prefer to feed on are considered. Fishes feeding on higher aquatic plants (commonly known as waterweeds), include grass carp, *Megalobrama amblycephala*, etc., and crustacean river crabs and freshwater shrimp also feed on some aquatic plants. Fish productivity is estimated according to the following Formula (9):Fp = Bhap × *P/B* × Ur ÷ Fcr × Ca × Cr(9)
where Fp is the fish productivity, Bhap is the biomass of higher aquatic plants, Ur is the utilization rate, Fcr is the feed conversion ratio, Ca is the culture area, and Cr is the coverage rate. Among them, the *P/B*, which is generally 1–2.5, is herein determined as 1.5; as the current waterweeds are still in the recovery period, the utilization rate of waterweeds is determined as 20%; and the feed conversion ratio, which is generally 80–120, is herein determined as 100. Accordingly, the productivity of fish, such as grass carp, feeding on spirodela polyrhiza and hornwort is 181,000 kg/year. For the purpose of clarification, as the current waterweed coverage rate of Shengjin Lake is less than 10%, they should be protected and recovered rather than being used by fish. Therefore, fish productivity in this regard is not included in the maximum carrying capacity.

In this way, the annual overall fish yield potential of Shengjin Lake, excluding waterweeds; that is, the annual maximum carrying capacity of the water body is (see Table 4): 150.0 + 129.5 + 31.0 + 28.0 = 3.385 million kg.

It must be noted that the fish production potential, i.e., the maximum fish carrying capacity, is not fixed and is mainly determined by the feed biomass in the water. In the next few years, the fish production potential in this regard may be increased if the existing fishery production mode is moderately adjusted, the coverage of aquatic vegetation, especially that of submerged plants, is recovered to the level of 1970–1980 through artificial measures, and measures such as increasing the natural proliferation and moderate artificial release of precious indigenous fish, such as *Pelteobagrus fulvidraco*, Culter, and mandarin fish, are taken. However, the fish production potential of filter-feeding fish, such as silver carp and bighead carp, is bound to decrease significantly. In general, excluding human disturbance, with the improvement of water quality, the fish production potential will not increase significantly, but the fish species will be further diversified, and the fishery output value will be upgraded.

## 5. Conclusions

According to the fishery development of Shengjin Lake in recent years, 40% of the fish production potential of the water body is converted into fish production, excluding the amount of water bird feeding and the amount that escape upstream and downstream. This means that the annual catch yield of Shengjin Lake at present should be 1.354 (3.385 × 0.4) million kg. The easy net access rate and catching rate of surface fish are higher, while the net access rate of demersal fish is lower, and the catching rate is slightly lower. These are the annual catching quotas for the fishery enterprises under the current management level. Of course, in recent years, it is not only forbidden to release herbivore fish, but the catching intensity of them also has to increase, so the catching quota could be slightly increased.

## 6. Discussion

The change in fish resources is an important driving factor for the evolution of lake ecosystems and the key to lake management and ecosystem restoration [35]. At present, the basic requirements for the management and prevention of inland water bodies in China are “protect water quality, give consideration to fishery, develop moderately, use sustainably”. At present, the problem the lake faces and urgently requires to be settled is that, on the basis of protecting water environments, implementing lake ecological restoration is necessary to effectively control the recession trend of lake fish resources and improve as well as maintain them.

The essence of the assessment of fish yield is to study the maximum amount of fishery products that can be formed by energy conversion and the utilization of different trophic creatures in water bodies, which is of great importance for the rational development and utilization of the natural bait resources of lakes and other water bodies [36]. According to the biomass, *P/B* ratio, bait ratio, and bait utilization rate of different aquatic creatures in a lake, the fishery capacity of fishes with different feed habits in the water can be estimated and, on this basis, can be used to guide the reasonable release, and optimized regulation of the fish structures in a lake. On the one hand, the fishery regulation of the lake, based on the evaluation results of the fish yield, can more accurately realize the multilevel utilization of bait resources to improve the fishery yield and output value; on the other hand, it is conducive to maintaining the stability and health of aquatic community structures and realizing the self-regulation and improvement of lake ecosystems [37]. Therefore, the investigation of aquatic creatures’ resources and the evaluation of fish yield should be the premise and implementation base for the fishery regulation and management work of propagation and releasing, as well as for the structural optimization of fish resources. It must be stated that, in future research, nonlinear models, such as convergent cross-mapping or OIF, would be a better choice, which can account for nonlinear variable interactions [9]. Furthermore, a global sensitivity and uncertainty analysis (GSUA) should be conducted to identify the key determinants of model indicator variability [38].

Due to the lack of proper planning and management of the lake, some problems of predatory fishing and unreasonable regional breeding arose in the fish resources of the lake, the living resources of the lake were destroyed, and the environmental water quality was decreased, which then affected the sustainable utilization and fishery capacity development of the fish resources of the lake. Fishing is one of the important functions of a lake; fish are the top regulators of the ecosystem, and so, after the control of exogenous pollution, the ecological restoration of eutrophic lakes should finally depend on the self-improvement and regulation of the structure of the ecosystem to realize the self-restoration of water bodies as well as the final balance and sustainability of health [1]. Therefore, on the premise of protecting the water environments of lakes, it will be an important means for the orderly regulation and management of fish resources to rationally locate lake fishery functions, reasonably plan fishery areas, determine the environmental capacity of lake fishery, and put forward the optimal fishery scale and fishery mode.

## Figures and Tables

**Figure 1 ijerph-19-08177-f001:**
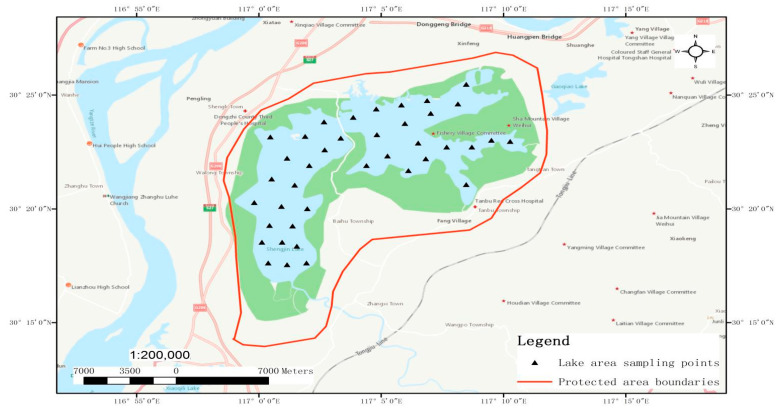
Distribution of field survey sites in Shengjin Lake.

**Figure 2 ijerph-19-08177-f002:**
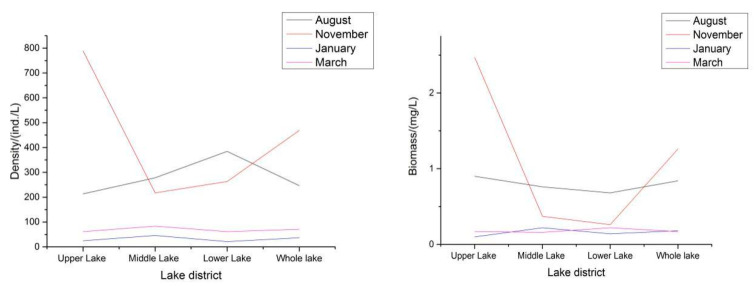
The trends in the zooplankton density and biomass in each lake area of Shengjin Lake.

**Figure 3 ijerph-19-08177-f003:**
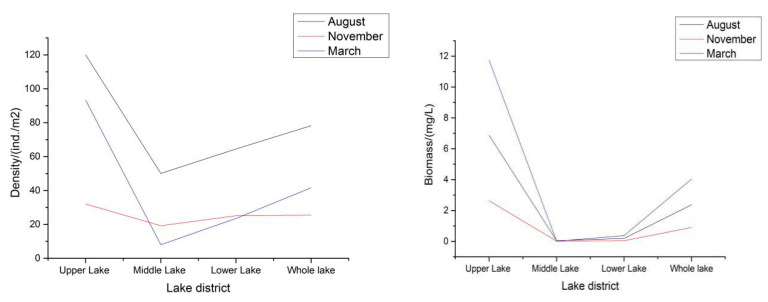
The trends in the density and biomass of benthos in each lake area of Shengjin Lake.

**Table 1 ijerph-19-08177-t001:** Distribution of the spawning grounds in Shengjin Lake.

SN	Site	Habitat	Species of Spawning Fish	Size (Length × Width, km)	Spawning Time
1	Yangetou–Landaochen	Marsh	*Spualiobarbus Curriculus*, *Megalobrama skolkovii*, *Xenocypris davidi*, etc.	6 × 0.5	June–August
2	Huangpen Gate–Babaizhang	Marsh	Crucian carp hybrid, *Culter alburnus*, *Odontobutis obscurus*, etc.	6 × 0.5	

**Table 2 ijerph-19-08177-t002:** Species, coverage, and biomass of higher aquatic vascular plants in Shengjin Lake.

Family	Name of Species	Coverage (%)	Biomass per m^2^ (Wet Weight) g
*Umbelliferae*	Wild carrot	80	486
	Shechuangzi	85	462
*Trapaceae*	Water chestnut	70	538
*Compositae*	*Artemisia selegensis*	75	398
*Potamogetonaceae*	Curled pondweed	45	356
*Haloragaceae*	Whorled water-milfoil	60	481
*Hydrocharitaceae*	*Hydrilla*	25	196
*Lemnaceae*	*Spirodela polyrhiza*	40	95
*Ceratophyllaceae*	Hornwort	35	237
*Poaceae*	*Zizania latifolia*	75	448
	Stinkgrass	90	406
*Cyperaceae*	*Carex thunbergii*	95	628

**Table 3 ijerph-19-08177-t003:** Species and dominance of dominant zooplankton species.

Dominant Species	Dominance			
	14.8	14.11	15.1	15.3
* **Rotifera** *				
*Keratella cochlearis*	-	0.106	-	0.1
*Asplachna priodonta*	0.175	0.023	-	0.104
*Polyarthra trigla*	0.159	0.308	0.2244	0.223
*Brachionus diversicornis*	0.347	0.083	-	-
*Filinia maio*	-	0.028	0.0323	0.035
*Brachionus calyciflorus*	0.045	-	0.2253	0.23
*Brachionus falcatus*	0.042	-	-	-
*Keratella valga*	0.034	-	-	-
*Trichocerca cylindrica*	0.021	-	-	-
*Rotoria tardigrada*	-	-	-	0.041
*Brachionus urceus*	-	-	-	0.055
*Brachionus angularis*	-	-	0.1594	-
*Notholon labis*	-	-	0.0587	-
* **Copepods** *				
*Thermocyclops hyalinus*	0.518	0.153	-	-
*Sinocalanus dorrii*	-	0.171	0.1167	-
*Mesocyclops leuckarti*	0.167	-	-	-
*Cyclops vicinus*	-	-	0.7268	-
* **Cladocera** *				
*Bosmina*	-	0.166	0.1016	-
*Diaphanosoma*	0.846	0.247	-	-
*Bosminopsis deitersi*	0.022	-	-	-
*Pleuroxus hamulatus*	-	-	0.2083	-

**Table 4 ijerph-19-08177-t004:** Estimation of current annual fish productivity of various feed resources in Shengjin Lake (ten thousand kilograms).

	Biomass	Utilization %	*P*/*B* Ratio	Bait Ratio	Fish Yield	Carrying Capacity	Fish Fed on Bait Resources	Ratio of Releasing	Regions
Phytoplankton	8.33 (mg/L)	30	300	100	150.0	150.0	Mainly of *Hypophthalmichthys molitrix* (silver carp)	55–60%	Open water
Zooplankter	2.59 (mg/L)	50	50	10	129.5	129.5	Mainly of bighead carp (spotted silver carp)	25–30%	Open water
Benthos	2.44 (g/m^2^)	25	5	20	31.0	31.0	Mainly of black carp, carp, crucian carp, and *Pelteobagrus fulvidraco*	10–20% Release *Xenocypris*, *Cyprinus carpio*, crucian carp, and *Pelteobagrus fulvidraco* rather than black carp	Shallow water area along the bank
Organic detritus		The fish yield is about 10–15% of that of filter-feeding fish			28.0	28.0	Mainly of *Xenocypris*, *Cyprinus carpio*, and crucian carp		
Higher aquatic plants	332 (g/m^2^)	20	1.5	100	18.1		Mainly of *Eriocheir sinensis*, *Ctenopharyngodon idellus*, and *Megalobrama amblycephala*	No release in the early period until the coverage rate of aquatic plant is greater than 30%	
Total					356.6	338.5	677.3		

## Data Availability

The data presented in this study are available on request from the corresponding author. The data are not publicly available due to government regulation. Publicly available datasets were analyzed in this study. This data can be found here: http://sjh.shidi.org/sjhbhq.html.

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
