# Peer review of "Analysis of the Ecological Carrying Capacity of Fish Resources in Shengjin Lake, Anhui Province, China"

_ijerph, 2022, doi:10.3390/ijerph19138177_

Round 1

Reviewer 1 Report

SPECIFIC COMMENTS 

The paper is potentially interesting and it may deal with eco-hydrological control/restoration of lakes to improve fishery, considering Ecological Carrying Capacity as indicator. The problem, however, is that the broad research question are badly posed because everything is just a very specific assessment of a very specific lake without any major finding of general applicability.

Also the paper is very poorly written with many syntax mistakes, e.g. see below this part in the abstract:

''Based on the background data of Shengjin Lake fishery ecosystem, this paper uses the bait base estimation method to evaluate the fish carrying capacity of Shengjin Lake, which provides a scientific basis for the implementation of fishery ecosystem management in Shengjin Lake National Nature Reserve. As the coverage of water grass in Shengjin Lake is less than 10%, it should be protected and restored, rather than used by fish, so the fish yield based on higher aquatic plants is not included in the maximum bearing capacity. ''

Many repetition are present and long sentences.

Also, visualization of results are non-existent... just tables is not enough. Maps and trends or distributions must be present. 

Carrying capacity should not be confused with abundance so some clarification should be made in that sense. 

Further comments are below on technical issues.   

GENERAL COMMENTS: 

(1) The ecological-environmental processes you investigate are largely non-linear. Non-linear models such as Convergent Cross Mapping or OIF (see Li & Convertino 2021 for instance) can account for variable non-linear interactions even without considering time delays (that are however quite important for example for your ecological dynamics, i.e. fish variability over space-time). These models are also able to capture spatial (ecology and environmental networks) variability to understand spatial dependencies that are really important for the variable considered. I am not sure how your study did consider these non-linearities and then if you can consider these aspects it would be quite relevant to truly map ecosystem dynamics (or otherwise stating this as a limitation of the study). The model you use is essentially linear and that may largely bias the results, e.g. when you link environmental and ecological variables. Spatial networks can reveal likely ecological and environmental mechanisms underpinning ecological emergence (e.g. hydrological processes). 

(2) To address the model/analysis Uncertainty-Sensitivity-Relevancy trilemma, global sensitivity and uncertainty analysis (GSUA) should be done to identify key determinants of model/analysis indicator variability (e.g. carryin capacity). The authors lack to perform a classical sensitivity analysis too and so it is really not clear what are the non-linear/synergistic drivers of model output/data (in relation to non-linear processes). GSUA considers non-linearity dependent on variable interactions (e.g. how different input factors are interlinked over space and time). See Pianosi et al. (2016) for an extensive discussion about this topic and how data should be used for GSUA using a simple variance-based approach. It is essentially looking into how much variability in contained in inputs for the variability of outputs (or event better into the co-predictability versus causality based on pdfs). 

(3) How indicators/predicted variables (i.e. carrying capacity) change over space is critical to understand site-/time-specific and universal (ecosystem invariant) shifts. Thus, indicator distributions can be analyzed as a function of the environmental variables (or predictors) considering joint probability distribution functions (pdfs), or average value and variance and looking into indicator variability as a function of predictor gradients (eco-env factors). Stability of ecological patterns over predictor gradients is important to quantify because that can define potential stable states over which the predicted variable is not changing. 

RECOMMENDATION:

I suggest to accept the paper after very Major Revision for sure. The paper is potentially interesting but must present more clear analyses of the of eco-environmental nexus and more general findings. I also think the writing is very poor. I think it is not clear what is really driving carrying capacity and how much uncertainty is around environmental controls. A fully probabilistic characterization would be interesting and important. 

REFERENCES:

Li J, Convertino M (2021) Temperature increase drives critical slowing down of fish ecosystems. PLoS ONE 16(10): e0246222. https://doi.org/10.1371/journal.pone.0246222

Pianosi et al. (2016)

Sensitivity analysis of environmental models: A systematic review with practical workflow

Environmental Modelling & Software

Volume 79, May 2016, Pages 214-232

Packages for GSUA 

- https://www.safetoolbox.info/info-and-documentation/ 

Author Response

Response to Reviewer 1 Comments.

Point 1:''Based on the background data of Shengjin Lake fishery ecosystem, this paper uses the bait base estimation method to evaluate the fish carrying capacity of Shengjin Lake, which provides a scientific basis for the implementation of fishery ecosystem management in Shengjin Lake National Nature Reserve. As the coverage of water grass in Shengjin Lake is less than 10%, it should be protected and restored, rather than used by fish, so the fish yield based on higher aquatic plants is not included in the maximum bearing capacity. ''

Response 1:  

The paper adopts the background data of Shengjin Lake fishery ecosystem and uses the bait based estimation method.The results show that: (1) the annual yield of silver carp fed on phytoplankton is 1.5 million kg; (2) the annual yield of bighead carp fed on zooplankton is 1.295 million kg; (3) the annual yield of benthos is 310,000 kg; (4) the annual yield of organic detritus is 280,000 kg; (5) As the coverage of water grass in Shengjin Lake is less than 10%, it should be protected and restored, rather than used by fish. In general, the annual maximum carrying capacity of fish in Shengjin Lake is 3.385 million kg except for water and grass.

Point 2:Many repetition are present and long sentences.

Response 2:

Long sentences in the abstract have been modified.

Some syntax errors have been corrected.

Point 3:Also, visualization of results are non-existent... just tables is not enough. Maps and trends or distributions must be present. 

Response 3:   

Figure 1 (Distribution of field survey sites in Shengjin Lake) is supplemented.

Maps and trend charts are supplemented.

Figure 2 (The trend of zooplankton density and biomass in each lake area of Shengjin Lake)is supplemented.

Figure 3 (The trend of density and biomass of benthos in each lake area of Shengjin Lake)is supplemented.

Point 4:Carrying capacity should not be confused with abundance so some clarification should be made in that sense.

Response 4:

Carrying capacity should not be confused with abundance. It has been clarified.

Point 5:The ecological-environmental processes you investigate are largely non-linear. Non-linear models such as Convergent Cross Mapping or OIF (see Li & Convertino 2021 for instance) can account for variable non-linear interactions even without considering time delays (that are however quite important for example for your ecological dynamics, i.e. fish variability over space-time). These models are also able to capture spatial (ecology and environmental networks) variability to understand spatial dependencies that are really important for the variable considered. I am not sure how your study did consider these non-linearities and then if you can consider these aspects it would be quite relevant to truly map ecosystem dynamics (or otherwise stating this as a limitation of the study). The model you use is essentially linear and that may largely bias the results, e.g. when you link environmental and ecological variables. Spatial networks can reveal likely ecological and environmental mechanisms underpinning ecological emergence (e.g. hydrological processes). 

Response 5:  

This paper adopts the bait based estimation method, and uses the linear model, which refers to  Hydrobiology of a Flooding Ecosystem, Lake Chenhu in Hanyang, HuBei, with Preliminary Estimation of its Potential Fishery Production Capacity(Liang Yanling et al.,1988). The estimation method of fish productivity potential in the document has been cited by Chinese scholars for many years to evaluate the fish productivity of lakes in the Yangtze River Basin in China. Of course, the reviewers' suggestions on non-linear models have opened our eyes and provided support for our future research.

Point 6:To address the model/analysis Uncertainty-Sensitivity-Relevancy trilemma, global sensitivity and uncertainty analysis (GSUA) should be done to identify key determinants of model/analysis indicator variability (e.g. carryin capacity). The authors lack to perform a classical sensitivity analysis too and so it is really not clear what are the non-linear/synergistic drivers of model output/data (in relation to non-linear processes). GSUA considers non-linearity dependent on variable interactions (e.g. how different input factors are interlinked over space and time). See Pianosi et al. (2016) for an extensive discussion about this topic and how data should be used for GSUA using a simple variance-based approach. It is essentially looking into how much variability in contained in inputs for the variability of outputs (or event better into the co-predictability versus causality based on pdfs).

Response 6:

We have not used the non-linearity reflecting the interaction of variables, so GSUA will be fully studied in the follow-up study.

Point 7:How indicators/predicted variables (i.e. carrying capacity) change over space is critical to understand site-/time-specific and universal (ecosystem invariant) shifts. Thus, indicator distributions can be analyzed as a function of the environmental variables (or predictors) considering joint probability distribution functions (pdfs), or average value and variance and looking into indicator variability as a function of predictor gradients (eco-env factors). Stability of ecological patterns over predictor gradients is important to quantify because that can define potential stable states over which the predicted variable is not changing. 

Response 7:

Thank you very much for your valuable comments. As for the analysis of index distribution as a function of environmental variables, we will give full consideration to it in subsequent studies. In this way, the thesis will be more logical.

Reviewer 2 Report

The submitted manuscript entitled "Analysis on Ecological Carrying Capacity of Fish Resources in Shengjin Lake, Anhui Province, China" cod. ijerph-1735362 by Zhang et al. it could be an interesting manuscript if structured better.

In particular, the introduction extensively mentions studies carried out by other scholars in the same field of investigation but at the end of the paragraph, it is not possible to understand the objectives of the work. Therefore I ask the authors to rewrite the introduction and highlight the objectives. The manuscript as a whole is divided into too many subparagraphs that break up the entire study, making the reading unpleasant. Therefore I ask the authors to rewrite paragraph 4 above all. The authors wrote the discussion paragraph including the conclusions. I prefer the conclusions paragraph separate from the discussions.

Author Response

Response to Reviewer 2 Comments.

In particular, the introduction extensively mentions studies carried out by other scholars in the same field of investigation but at the end of the paragraph, it is not possible to understand the objectives of the work. Therefore I ask the authors to rewrite the introduction and highlight the objectives. The manuscript as a whole is divided into too many subparagraphs that break up the entire study, making the reading unpleasant. Therefore I ask the authors to rewrite paragraph 4 above all. The authors wrote the discussion paragraph including the conclusions. I prefer the conclusions paragraph separate from the discussions.

Response 1:We have slightly adjusted the structure of the thesis.

Response 2:The fourth paragraph of the introduction is rewritten to highlight the research objectives of the thesis.

“As one of the lakes flowing into the middle and lower reaches of Yangtze River, Shengjin Lake is a major overwintering place and ceasing place for East Asian and Australian migratory water birds. It has a significant ecological status and international influence, and was approved as National Natural Reserve in 1997. The water level of Shengjin Lake decreases in autumn and winter and increases in spring and summer with the change of seasonal precipitation, and it has a higher fish production capacity [32,33]. However, with the increasing environmental constraint in recent years, the fishery development space of Shengjin Lake has greatly shrunk, so that the fish production management method requires urgent transformation and upgrading. In the face of the new situation, and according to the demands of healthy ecologic system of large water surface and fishery development, it has become the realistic problems urgently needed to be solved on how to conduct regulatory activities of fish production in Shengjin Lake and how to promote the harmonious development of ecology, production and life of the water area. This paper follows the principle of sustainable utilization of fish resources and uses the background data of Shengjin Lake fishery ecosystem to evaluate the fish production potential of Shengjin Lake. The research objective is to scientifically set sustainable fishing quotas to optimize fishing.”

Response 3:Separate the conclusion from the discussion.

Response 4:In response to the research objective of the paper in paragraph 4, the conclusion is further clarified.

“Conclusion

According to the fishery development of Shengjin Lake in recent years, 40% of the fish production potential of the water body is converted into fish production, excluding the amount of water bird feeding and escape from upstream and downstream.It means the annual catch yield of Shengjin Lake at present should be 1.354 (3.385 ×0.4) million kg. In which, the easy net-access rate and catching rate of surface fish are higher, while the net-access rate of demersal fish is lower and the catching rate is slightly lower. These are the annual catching quota for the fishery enterprises under the current management level. Of course, in recent years, it is not only forbidden to release herbivore fish, but also have to increase the catching intensity of them, so that the catching quota could be slightly increased.”

Furthermore,some irrelevant literature has been deleted.

Articles 15, 25, 28, 31, 32 and 35 in the manuscript returned on June 19, 2022 were deleted.

Articles 9 and 38 in today's manuscript are added.

Round 2

Reviewer 1 Report

I am ok with the paper but I am feeling like my comments have been only tangentially addressed in a written response, or avoided completely.... e.g. of course carrying capacity is related to abundance, specifically as upper limit within a certain ecological fitness. Carrying capacity refers to the maximum abundance of a species that can be sustained within a given area of habitat . Theoretically when an ideal population is at equilibrium with the carrying capacity of its environment, the birth and death rates are equal, and size of the population does not change. Stochastically, this means that fluctuations sum up to the same average value. 

Other comments have been postponed to further studies despite data (and not model) are non-linear... so adopting a linear model is not a proper choice... 

In conclusion I am ok to accept but I believe certain clarification must be made.

Author Response

Dear reviewer,

Thank you very much.Your comments reflect your tolerance for our research work and your help to us.

Here is our response.

Point 1:

Of course carrying capacity is related to abundance, specifically as upper limit within a certain ecological fitness. Carrying capacity refers to the maximum abundance of a species that can be sustained within a given area of habitat .

Response 1:We agree with your point of view, which has been reflected in the latest revised manuscript.

Point 2:Other comments have been postponed to further studies despite data (and not model) are non-linear... so adopting a linear model is not a proper choice...

Response 2:

As you commented, carrying capacity is the upper limit within a certain ecological fitness. We use a linear model to calculate the maximum abundance of fish carrying capacity in Shengjin Lake.Now we are also aware of the limitations of doing so. In the follow-up study, we will try to use the non-linear model to calculate this carrying capacity.

Thank you again.

That's all.

Kind regards.

Reviewer 2 Report

Dear Authors,

why insert the conclusions paragraph before the discussions?

The conclusion of a research paper restates the research problem, summarizes your arguments or findings, and discusses the implications.

Author Response

Dear reviewer,

Thank you very much.

Here is our reply.

Point:

Why insert the conclusions paragraph before the discussions?

Response :

First, the conclusion responds to the research goal of this paper in the introduction, and obtains the current sustainable fishing quota of Shengjin Lake. Therefore, it is placed immediately after the analysis results. Second, the discussion part focuses on how to carry out the dynamic management according to the changes of fishery resources in Shengjin Lake under the constraints of current carrying capacity.Third, the discussion part involves different research methods and perspectives, how to make the demonstration process more logical.

That's all.

Kind regards.

This manuscript is a resubmission of an earlier submission. The following is a list of the peer review reports and author responses from that submission.